

# Estimating the saturation vapor pressures of isoprene oxidation products C$_5$H$_{12}$O$_6$ and C$_5$H$_{10}$O$_6$ using COSMO-RS

Theo Kurtén[1], Noora Hyttinen[1], Emma L. D'Ambro,[2,3] Joel Thornton,[2,3] Nønne L. Prisle[4]

[1]University of Helsinki, Department of Chemistry and Institute for Atmospheric and Earth System Research (INAR), Helsinki, FI-00014, Finland
[2]Department of Chemistry, University of Washington, Seattle, Washington 98195, United States
[3]Department of Atmospheric Sciences, University of Washington, Seattle, Washington 98195, United States
[4]University of Oulu, Nano and Molecular Systems Research Unit, P.O. Box 3000, 90014, University of Oulu, Oulu, Finland

*Correspondence to*: Theo Kurtén (theo.kurten@helsinki.fi)

**Abstract.** We have used COSMO-RS (the COnductor-like Screening MOdel for Real Solvents), as implemented in the COSMOTherm program, to compute the saturation vapor pressures at 298 K of two photo-oxidation products of isoprene: the dihydroxy dihydroperoxide C$_5$H$_{12}$O$_6$, and the dihydroperoxy hydroxy aldehyde, C$_5$H$_{10}$O$_6$. The predicted saturation vapor pressures were significantly higher (by up to a factor of 1000) than recent experimental results, very likely due to the overestimation of the effects of intramolecular hydrogen bonds, which tend to increase saturation vapor pressures by stabilizing molecules in the gas phase relative to the liquid. Modifying the hydrogen bond enthalpy parameter used by COSMOTherm can improve the agreement with experimental results – however the optimal parameter value is likely to be system-specific. Alternatively, vapor pressure predictions can be substantially improved (to within a factor of 10 of the experimental values for the two systems studied here) by selecting only conformers with a minimum number of intramolecular hydrogen bonds. The computed saturation vapor pressures were very sensitive to the details of the conformational sampling approach, with the default scheme implemented in the COSMOconf program proving insufficient for the task, for example by predicting significant differences between enantiomers, which should have identical physical properties. Even after exhaustive conformational sampling, COSMOTherm predicts significant differences in saturation vapor pressures between both structural isomers and diastereomers. For C$_5$H$_{12}$O$_6$, predicted differences in p$_{sat}$ between structural isomers are up to two orders of magnitude, and differences between stereoisomers up to a factor of 20 - though these differences are very likely exaggerated by the overestimation of the effect of intramolecular H-bonds. For C$_5$H$_{10}$O$_6$, predicted differences between (stereo)isomers are below a factor of 3. In future studies of saturation vapor pressures of polyfunctional atmospheric oxidation products using COSMOTherm, we recommend first performing thorough conformational sampling, and subsequently selecting conformers with a minimal number of intramolecular H-bonds.



## 1. Introduction

Atmospheric aerosol particles play a key role in regulating the Earth's climate, and are responsible for most air pollution-related mortality (Pachauri and Meyer, 2014; Brauer et al., 2016). A large fraction of these particles consists of secondary organic aerosol (SOA) material formed by oxidation reactions (Jimenez et al., 2009). Until recently, both the amount and the condensability of SOA were severely underestimated by atmospheric chemistry models, leading to large gaps in the understanding of atmospheric aerosols (Zhang et al, 2014; Pierce et al., 2011). A combination of field studies, laboratory work and modelling has shown that a large part of the missing SOA may be explained by peroxyradical ($RO_2$) autoxidation reactions, which rapidly convert hydrocarbons into highly oxidized multifunctional compounds (HOM; Ehn et al., 2014; Bianci et al., 2018). Unlike the sequential oxidation reactions which dominate the chemistry of simple "textbook" species with few functional groups, autoxidation only needs a single initial hydrocarbon – oxidant reaction: the subsequent cascade of $RO_2$ hydrogen shift (H-shift) and $O_2$ addition reactions can add up to 10 oxygen atoms to a hydrocarbon backbone without needing any additional low-concentration oxidants such as OH, $O_3$ or $NO_3$ (Ehn et al., 2014).

HOM compounds formed by $RO_2$ H-shifts likely contain multiple hydroperoxide and/or peroxy acid functional groups, in addition to the alcohol, carbonyl and carboxylic acid groups found in "conventional" atmospheric oxidation products (Bianci et al., 2018). This creates a challenge for evaluating the physical properties determining their atmospheric behaviour and impact, such as pure-compound saturation vapour pressures ($p_{sat}$), solubilities (in various atmospherically relevant condensed phases) or activities. Until very recently, there were essentially no data on the volatility of polyfunctional peroxide compounds (defined here as compounds containing more than one hydroperoxide or peroxy acid group, and at least one other non-alkyl functional group). Due to this limitation, empirical group contribution methods such as the generally successful Nannoolal (Nannoolal et al., 2008), SIMPOL (Pankow and Asher, 2008) or EVAPORATION (Compernolle et al., 2011) approaches cannot be expected to give reliable predictions for the properties of HOM compounds. In addition to this specific issue related to polyfunctional peroxides, there are also indications that group contribution methods may generally underestimate the volatility of highly functionalized compounds (Meara et al., 2014; Valorso et al., 2011). Both these issues may be related to limitations in the ability of these methods to describe intramolecular interactions, especially intramolecular hydrogen bonds. These tend to stabilize molecules in the gas phase relative to the condensed phase, and thus lead to higher vapor pressures, as illustrated for example by the higher vapor pressure of 1,2-benzenediol (cathecol), which can form intramolecular H-bonds, compared to 1,4-benzenediol (hydroquinone), which cannot (Chen et al., 2006). While intramolecular H-bonding is in principle described by cross terms in many of the group contribution methods, lack of data has so far prevented the development of appropriate cross terms to describe for example H-bonds between multiple OOH groups.

Estimation methods based on quantum chemistry, most notably COSMO-RS (COnductor-like Screening MOdel for Real Solvents; Eckert and Klamt, 2002), as implemented in the COSMOTherm program (COSMOTherm, 2018), require both fewer and less system-specific empirical parameters, and could thus be expected to be more accurate for compound classes for



which there are limited measurement data. COSMOTherm has recently been applied to estimate both saturation vapor pressures, Henry's law constants and partitioning coefficients for atmospherically relevant oxidized organic molecules, including some potential autoxidation products (Wang et al., 2017; Wang et al., 2015; Wania et al., 2014; Kurtén et al., 2016). Compared to group contribution methods, COSMOTherm has generally been found to predict significantly higher saturation

vapor pressures and lower gas-aqueous partitioning coefficients (Wang et al., 2017) for polyfunctional oxidized species. In contrast, the disagreement between gas – organic phase partitioning coefficients is considerably smaller, as the discrepancies in predicted saturation vapor pressures and activity coefficients in water-insoluble organic phases partially cancel each other (Wania et al., 2014). It is unclear whether the difference in predicted saturation vapor pressures is mainly due to an overestimation by COSMOTherm, an underestimation by the group contribution methods, or a combination of both.

COSMOTherm is in principle able to account for intramolecular hydrogen bonding interactions between any types of functional groups, as well as for the related conformational complexity of polyfunctional species. For treating large sets of compounds, the COSMOconf module for conformer sampling (COSMOconf, 2017) is an attractive add-on to COSMOTherm, as it allows for the rapid and automated prediction of properties using for example SMILES strings as input. However, due to the general lack of measurement data on low-volatility species (Bilde et al., 2015), neither the accuracy of the absolute $p_{sat}$ (or

partitioning coefficient) predictions, or the performance of the default approaches for conformational sampling implemented in COSMOconf, have been evaluated for polyfunctional peroxide species. In this study we attempt to perform such evaluations, using two recently reported experimental saturation vapor pressure results as benchmarks.

       D'Ambro et al. (2017) recently used thermal desorption measurements, combined with an iodide-based high-resolution time-of-flight chemical ionization mass spectrometer (HR-ToF-CIMS) coupled to the Filter Inlet for Gases and

AEROsols (FIGAERO; Lopez-Hilfiker et al., 2014) to estimate the saturation vapor pressures of several products of isoprene photo-oxidation. Isoprene, $C_5H_8$, is the main hydrocarbon emitted to the air from vegetation, with a global mass flux estimated as 500 TgC/yr (Guenther *et al.*, 2012). While the relative SOA yield from isoprene is low compared to larger hydrocarbons such as monoterpenes, the large emissions make isoprene one of the most important SOA precursors on a global scale (Wennberg et al., 2018). Two of the strongest signals detected in the particle phase from isoprene + OH (for both high and low

$NO_x$ conditions) are $C_5H_{12}O_6$ and $C_5H_{10}O_6$ (Liu et al., 2016), tentatively identified as a dihydroxy dihydroperoxide (denoted ISOP(OOH)$_2$) and a dihydroperoxy hydroxy aldehyde, with estimated saturation vapor pressures at 298 K of $9.87\times10^{-7}$ mbar and $6.70\times10^{-6}$ mbar, respectively (D'Ambro et al., 2017). While technically not a "pure" autoxidation product, as its formation mechanism likely involves two OH oxidation steps (see D'Ambro et al., 2017 and Wennberg et al., 2018 for details), ISOP(OOH)$_2$ shares many features of the proposed autoxidation products of larger alkenes such as monoterpenes: it has an

O:C ratio above one, contains two OOH groups and two other functional groups, and potentially forms up four intramolecular hydrogen bonds. At the same time, ISOP(OOH)$_2$ is small enough that the uncertainty related to its chemical structure is relatively small (i.e. $C_5H_{12}O_6$ is highly likely to be a dihydroxy dihydroperoxide, though multiple structural isomers may coexist), and also the computational cost of treating all possible structural isomers and conformers of ISOP(OOH)$_2$ using



quantum chemical methods is not prohibitive. To verify that our results are not specific to this particular molecule, we also performed calculations on the dihydroperoxy hydroxy aldehyde $C_5H_{10}O_6$.

## 2. Computational Methods

5  The saturation vapor pressures of the different $ISOP(OOH)_2$ structural isomers and stereoisomers were estimated using the COSMOTherm program (Version C3.0, release 18), with the BP_TZVPD_FINE_18 parametrization. The input files for COSMOTherm were generated using the COSMOconf program, which uses multiple steps to generate the input files corresponding to a representative set of unique conformers. The program includes various conformer sampling and clustering methods (including elimination of similar structures), and various quantum chemical methods for geometry optimizations and

10 single-point calculations in both the gas and the liquid phases. For the quantum chemical calculations, COSMOconf uses the TURBOMOLE program package (version 7.11; Turbomole, 2018) and the Becke-Perdew (BP) density functional (Becke, 1988; Perdew, 1986).

 We tested four different calculation schemes for generating and selecting the conformers for the final vapor pressure

15 calculations:

- *Systematic*, where the conformer sampling is done systematically using the Spartan'14 program (Wavefun, inc, 2014) and the MMFF94 force field, rotating over all dihedral angles in 120 degree increments. Note that no solvent model was included in the Spartan calculations. The full set of conformers from Spartan'14 was then used as input

20  for COSMOconf.

- *Best*, where the conformer search, using the Balloon program (Vainio et al., 2007; Puranen et al., 2010) implemented in COSMOconf, is started from a geometry that corresponds to the lowest COSMO-RS energy conformer calculated at the BP/def2-TZVPD//BP/def-TZVP level of theory. For this calculation scheme, all conformers were first generated using the systematic conformer sampling algorithm in the Spartan'14 program as

25  described above, and subsequently optimized at the BP/def2-TZVPD//BP/def-TZVP level of theory (including COSMO solvation) using TURBOMOLE 7.11.

- *Worst*, identical to *Best*, except with the COSMOconf search started from the highest-energy conformer found in the systematic sampling. The purpose of the *Best/Worst* schemes is to assess how much the vapor pressures depend on the arbitrary starting geometry provided to COSMOconf.

30 - *SMILES*, where only COSMOconf was used to generate conformers, using the default settings of the program, and creating the starting geometry from a SMILES string. This corresponds to results that would be obtained by a "casual" user, or in automated runs of large numbers of compounds.



The detailed settings for each conformational sampling scheme are shown in Table S1 of the Supplementary Information, together with additional computational details. The gas phase calculations were performed similarly in all four conformational sampling schemes: gas phase geometry optimizations at the BP/def-TZVP level of theory, and single-point

calculations at the BP/def2-TZVPD level of theory. In addition, COSMOconf maps the gas phase conformers to the COSMO conformers to ensure that each COSMO conformer has a single corresponding gas phase conformer. In the mapping, each gas phase conformer is assigned to the geometrically closest COSMO conformer. If more than one gas phase conformer corresponds to the same COSMO conformer, the gas phase conformer with the lowest energy is chosen. Single-point gas phase energies on the COSMO geometries are used for the COSMO conformers left without a corresponding gas phase

conformer after the mapping. The vapor pressure calculations were finally performed using 10 conformers in the *SMILES* scheme, and 100 conformers in the other schemes. The choice of 100 conformers was governed by a compromise: on one hand the number is large enough that the saturation vapor pressure results are more or less "converged" (at least to within a few tens of percent, see Sect. 3.1.2 for a test case) with respect to the number of conformers, on the other hand the number is small enough that the COSMOTherm calculations can be done within hours rather than days. (Also, COSMOTherm may sometimes

run out of memory if the number of conformers is much larger than 100.) We note that for many applications, a much smaller number of conformers may be sufficient, provided that the initial conformational sampling is thorough.

## 3. Results and discussion

### 3.1. Vapor pressure of $C_5H_{12}O_6$

As illustrated in Fig. 1, there are in principle six different structural isomers of the dihydroxy dihydroperoxide $C_5H_{12}O_6$ (assuming that the carbon backbone is identical to that of the parent isoprene). Each structural isomer has two chiral centers (carbon atoms with four non-equivalent substituents), giving rise to two pairs of enantiomeric structures. Based on the known mechanism of isoprene photo-oxidation, isomer 1 is likely to be the dominant form found in the atmosphere, with possible minor contributions from isomers 2, 3 and 4. In the recent review by Wennberg et al. (2018), the OH addition channels leading

to isomer 1 (from the two main structural isomers of isoprene hydroxy hydroperoxide, ISOPOOH, after several reaction steps) are estimated to have yields of 95%, compared to only 5% for the channels leading to other isomers. However, it should be noted that the products of the minor addition channels may have larger yields for the reactions actually forming the dihydroxy hydroperoxides, which are themselves only relatively minor products of the overall isoprene photo-oxidation process. The precise distribution of the different atmospherically relevant isomers is impossible to accurately estimate at the moment as

many of the key branching ratios are highly uncertain.



The computed vapor pressures for different structural isomers and stereoisomers of $C_5H_{12}O_6$ are shown in Table 1. For the first three structural isomers (isomers 1, 2 and 3), all four stereoisomers were studied. This provides an addition test
for the configurational sampling methods, as the physical pure-compound properties of enantiomeric pairs (i.e. SS/RR or SR/RS stereoisomers) should be identical. For the three remaining isomers (4, 5 and 6), only one of each enantiomeric pair was included for computational reasons.

Three main conclusions can be drawn from the data in Table 1. First, only the *Systematic* conformational sampling scheme results in enantiomers (mirror image pairs, i.e. RR/SS and RS/SR stereoisomers) having identical (or almost identical)
saturation vapor pressures. This is illustrated in Fig. 2 for the case of isomer 1 (see Fig. S1-S3 in the Supplementary Information for figures illustrating $p_{sat}$ for the other isomers). The other conformational sampling schemes are thus incomplete, as they lead to physically inconsistent results. The *SMILES* scheme, which corresponds to using the default conformational sampling settings in the COSMOconf program, leads to especially inconsistent results with differences of up to two orders of magnitude compared to the systematic conformational sampling. The difference between *Best* and *Worst* schemes can also be up to two
orders of magnitude, illustrating a worrying dependence of the COSMOconf results on the arbitrary input structure even when a conformational sampling approach much more thorough than the default setting is used.

Second, the computed saturation vapor pressures are significantly higher than the measurement results. The values computed using the physically consistent *Systematic* scheme for the atmospherically relevant isomers (1, 2, 3 and 4) are between 25 and 1550 times higher than the experimental value. This discrepancy is larger than that reported recently for
polyethylene glycol (PEG) vapor pressures (Krieger et al., 2018), where COSMOTherm was found to systematically overestimate the saturation vapor pressures of the first five PEG oligomers by a factor of 3-40. The larger disagreement may be related to the fact that the smallest PEG oligomers typically form one and the larger oligomers two intramolecular H-bonds, while $ISOP(OOH)_2$ can form four. In many cases, the incomplete and inconsistent conformational sampling schemes (*SMILES*, *Best* or *Worst*) actually lead to values closer to the experimental one – this is due to these schemes missing many of the lowest-
energy conformers, which typically contain close to the maximum number (in this case 4) of intramolecular hydrogen bonds. Conformers with a smaller number of internal H-bonds typically lead to lower vapor pressures (as discussed in detail below) – supporting the hypothesis that the general overestimation of $p_{sat}$ is due to an overestimation of the effects of internal H-bonds.

Third, COSMOTherm predicts surprisingly large differences in vapor pressures both between structural isomers, and between different diastereomers (non-enantiomeric stereoisomers) of the same structural isomer, as illustrated in Fig. 3. The
differences between diastereomers can be up to a factor of 20 even when using the *Systematic* conformer sampling scheme. For the structural isomers, a general trend can be discerned: isomers with OOH groups adjacent to each other (isomers 1, 3 and 4) tend to have lower vapor pressures than isomers with OOH groups further from each other (isomers 2, 4 and 6). This is likely caused by somewhat less efficient intramolecular H-bonding in the former set of isomers due to steric strain. The differences between diastereomers is similarly very likely related to subtle differences in H-bonding patterns. For example,





the lowest-energy conformers of different diastereomers typically look quite different– see Fig. 4 for an example where the lowest-energy conformers of two diastereomers have different numbers of intramolecular H-bonds. Since COSMOTherm seems to overestimate the effect of intramolecular H-bonding, it is likely that also the differences between structural isomers and diastereomers are exaggerated. However, even if the true differences between diastereomers and/or structural isomers are

significantly smaller than predicted by COSMOTherm, they may still be large enough to matter for atmospheric applications. Even differences on the order of some tens of percent between saturation vapor pressures of diastereomers would lead to different behaviour with respect to condensation. Since different sources (including both different biogeochemical primary sources, and different source reactions occurring in the atmosphere) can produce different distributions of stereoisomers, this could lead to seemingly inexplicable differences in behaviour between "identical" chemical species. However, it should be

noted that the thermal desorption profiles of these compounds (D'Ambro et al., 2017) appear Gaussian-like, or in other words, do not display bi- or multimodal desorption peaks which would indicate volatility differences between isomers of ISOP(OOH)$_2$. We therefore conclude that either the isoprene photo-oxidation leads to only one specific (stereo)isomer of ISOP(OOH)$_2$, or, more likely, that the differences in volatilities between the formed (stereo)isomers are too small to be measurable with the FIGAERO instrument (i.e. less than an order of magnitude).

### 3.1.1 Activity coefficients in water and water-insoluble organic matter

One potential reason for the difference between predicted and measured saturation vapor pressures is that the saturation vapour pressures reported by D'Ambro et al. (2017) are obtained by thermal desorption from aerosol particles

containing a mixture of many different isoprene oxidation products, adsorbed on top of a solid ammonium sulfate core. If the activity coefficient of ISOP(OOH)$_2$ in this mixture strongly deviates from unity, then the measured vapor pressure value (in effect, the equilibrium vapor pressure of ISOP(OOH)$_2$ over the mixture; see e.g. Bilde et al, 2015, for a detailed discussion) could be very different from the pure-compound $p_{sat}$. To test whether this could be the case, we computed the activity coefficients at infinite dilution of the different ISOP(OOH)$_2$ isomers in an organic phase consisting of molecule "B" proposed

by Kalberer et al. (2004) as a good model for representing atmospheric water-insoluble organic matter (WIOM). We also compute activity coefficients in water for comparison, although these are unlikely to be relevant to the experiments by D'Ambro et al. due to the use of solid ammonium sulfate seed and moderate humidity (RH = 50%). The calculations were done for the 100 best conformers, as selected by the *Systematic* scheme described above. The results are given in Table 2. The activity coefficients of the different ISOP(OOH)$_2$ isomers vary between 1.37 and 2.90 in pure water, and between 6.97 and

16.03 in the model WIOM phase. This suggests that the measured effective $p_{sat}$ might possibly be up to one order of magnitude higher than the true pure-compound $p_{sat}$, but is very unlikely to be lower than the true value. Differences between the measured





values and true pure-compound saturation vapor pressures can thus not explain the discrepancy between COSMOTherm results and measurements.

### 3.1.2 Sensitivity of $p_{sat}$ to H-bonding Parameters

We next tested the effect on the predicted $p_{sat}$ values of various adjustments to the treatment of H-bonds by COSMOTherm. First, we examined switching **inter**molecular H-bonding off entirely in the COSMO-RS treatment using the "nohb" setting. (Note that **intra**molecular H-bonds cannot be similarly switched off as they are implicitly included in the quantum chemical energy calculations.) This resulted in a dramatic increase of the predicted saturation vapor pressures, by

between a factor of 10 and 180 for all studied isomers using the 100 best conformers selected by the *Systematic* scheme described above). This is reasonable, as H-bonds are the strongest intermolecular interactions between ISOP(OOH)$_2$ molecules – ignoring them thus leads to significantly weaker intermolecular binding, and higher saturation vapor pressures. While the effect of intramolecular H-bonds cannot be similarly switched off, the quantum chemical treatment of H-bonding can in principle be slightly improved by recomputing the gas-phase energies at a higher level of theory, and using these higher-level

energies in the Boltzmann weighting of conformers. We tested this approach by recomputing the electronic energies (at the BP/def-TZVP geometries) for the gas-phase conformers corresponding to the ten lowest-energy COSMO conformers of R,S enantiomer of isomer 1 at the CCSD(T)-F12a/VDZ-F12 level (Adler et al., 2007; Werner et al., 2011) using the Molpro 2015.1 program (Werner et al., 2012). Of these ten gas-phase conformers, nine contained (based on visual inspection) four intramolecular H-bonds, and one contained three. The relative electronic energy of the conformer with only three H-bonds

(compared to the lowest-energy conformer) was +2.5 kcal/mol at the BP/def2-TZVPD//BP/def-TZVP level, but only +1.2 kcal/mol at the CCSD(T)-F12a/VDZ-F12//BP/def-TZVP level – suggesting that the BP method overestimates the favourability of intramolecular H-bonds. The vapor pressure computed for the set of 10 conformers accordingly decreased by a factor of 2 when the higher-level energies were used for weighting the gas-phase conformers. While this change is in the right direction, it is insufficient to reconcile predicted $p_{sat}$ values with experiments, likely because the overestimation of intramolecular H-

bonds is still present in the DFT-generated input files, despite the improved gas-phase conformer weighting. In principle, the problem might be resolved by generating the entire input files at a higher level of theory (for example, some variant of coupled cluster theory) – but this would require an entirely new parameterization for COSMO-RS, and also result in significantly higher computational costs, as well as much steeper scaling of the cost with respect to system size.

The newest release of COSMOTherm (version 18) provides some additional tools for adjusting the treatment of

(intermolecular) H-bonds. Specifically, both enthalpic and entropic H-bonding parameters, as well as their temperature sensitivities, can be scaled. In principle this corresponds to four different parameters (labelled c0, c1, s0 and s1), but one of them (s1) is set to zero, leading in practice to three adjustable parameters. Test calculations (see Figure S4 in the Supplementary Information file) indicate that for the ISOP(OOH)$_2$ saturation vapor pressures at 298 K, varying the c0 and s0 parameters had



relatively small effects, while varying the c1 parameter (corresponding to the H-bonding enthalpy) had a large effect. We therefore tested the effect of varying c1 between 0.6 and 1.8 for all ISOP(OOH)$_2$ isomers, using the 100 conformers from the *Systematic* sampling scheme (see Figure S5 in the Supplementary information file for results). In order to match the experimental value of about $1\times10^{-6}$ mbar solely by scaling the c1 parameter, scaling factors of 1.61 and 1.36 would be needed

for the SS/RR and SR/RS enantiomer pairs of structural isomer 1, respectively.

Finally, we tested the effect on saturation vapor pressures of selecting only conformers with less than a certain number of full or partial intramolecular H-bonds, as identified by the COSMOTherm program (see the Supplementary Information file for details). In order to ensure a sufficient number of conformers also for calculations using the more stringent selection criteria (e.g. choosing only conformers with one H-bond), we first performed a new conformational sampling for the S,S and S,R

stereoisomers of structural isomer 1, using the *Systematic* approach, but keeping 500 rather than 100 conformers in the overall conformer set. Next, saturation vapor pressures were computed for the two stereoisomers, but picking only conformers with less than $n+m$ intramolecular H-bonds, where $n$ refers to full bonds and varied from 1 to 4, and $m$ refers to partial bonds, and varied from 0 to 3, such that $n + m \leq 4$. (There were no conformers with zero full H-bonds for ISOP(OOH)$_2$.) The results are shown in Fig. 5 (data are shown both for the original set of 100 conformers in total, and for the larger set of 500 conformers).

First, we can see that using 500 rather than 100 conformers results in a slight decrease in p$_{sat}$ even before the number of H-bonds are restricted; from $1.39\times10^{-3}$ to $1.09\times10^{-3}$ mbar for the S,S stereoisomer, and from $8.59\times10^{-5}$ to $6.65\times10^{-5}$ mbar for the S,R stereoisomer. Restricting the number of H-bonds has a dramatic effect on the saturation vapor pressure especially for the S,S stereoisomer, which had a much higher vapor pressure than the S,R isomer when all conformers were included. While there is little difference between the $n=4$ and $n=3$ cases for either isomer, restricting the number of full H-bonds to 2 results in

a factor of 30 decrease in p$_{sat}$ for the S,S isomer, but little difference for the S,R isomer. Restricting $n$ to 1 results in a further decrease of a factor of 3 for the S,S case, and a factor of 11 for the S,R case. The number of partial H-bonds has only a minor effect on the saturation vapor pressure. For the smaller values of $n$, the importance of sampling a sufficient number of conformers becomes evident, as the values corresponding to the sets containing 500 and 100 conformers overall diverge significantly. The reason for this is that in the latter case, especially the $n=1$ restriction leads to only <10 conformers being

included in the p$_{sat}$ calculations.

The p$_{sat}$ values obtained with *n=1, m=0* ($1.30\times10^{-5}$ and $5.30\times10^{-6}$ mbar for the S,S and S,R stereoisomers, respectively, using the overall set of 500 conformers) are both much closer to each other, and closer to the experimental value (around $10^{-6}$ mbar), than the p$_{sat}$ values obtained using the full set of conformers. This confirms our hypothesis that the overestimation of vapor pressures by COSMOTherm, as well as the exaggerated differences between isomers, is very likely

caused by an overestimation of the effects of intramolecular H-bonding. The remaining discrepancy of about a factor of 5-10 between the *n=1,m=0* p$_{sat}$ values and experimental results would suggest that the single remaining intramolecular H-bond causes an overestimation in p$_{sat}$ of about a factor of 5. This is remarkably consistent with the data both in Table 1, and in Krieger et al (2018), all of which could be reasonably well explained (given all the other error sources in both the computed



values and the experimental measurements) with an overestimation in $p_{sat}$ of roughly a factor of 5 for every strong intramolecular H-bond.

### 3.2. Vapour Pressure of $C_5H_{10}O_6$.

We computed saturation vapor pressures for the three structural isomers of $C_5H_{10}O_6$ that are likely to be atmospherically important (see Fig. 6). For each structural isomer, one of each enantiomeric pair was selected (as shown in Fig. 6). The systematic conformer sampling scheme was used for the calculations, but with the last two cut-offs of 150 and 100 conformers (see Table S1 in the Supplementary Information) replaced by 600 and 500, respectively. The best 100 conformers were then used in the $p_{sat}$ calculation. Activities in water and a WIOM phase represented by molecule "B" from

Kalberer et al (2004) were also computed. The results are given in Table 3. Analogous to $C_5H_{12}O_6$, the predicted saturation vapour pressures are around 50 -150 times larger than the experimental value of $6.70 \times 10^{-6}$ mbar. The upper limit of this range is lower than for $C_5H_{12}O_6$, possibly due to $C_5H_{10}O_6$ having one less H-bond donor, and thus one fewer intramolecular H-bond. The overestimation is again remarkably consistent with an error of roughly a factor of 5 per intramolecular H-bond (as $5^3 = 125$). The smaller number of H-bonds may also be the reason for the smaller differences between structural isomers and

diastereomers (which are all less than a factor of 3), as well as the on average higher activity coefficients in water, and lower activity coefficients in the WIOM phase, compared to $C_5H_{12}O_6$. Another reason for the smaller differences may be that all three studied isomers have OOH groups on adjacent C atoms, possibly leading to somewhat weaker intramolecular H-bonding (and thus lower $p_{sat}$) due to steric strain, as discussed in section 3.1.

      We also tested the effect of setting the H-bonding enthalpy parameter c1 to either 1.61 or 1.36 (the values needed to

match experiments for the two diastereomers of isomer 1 of $C_5H_{12}O_6$), as well as of restricting the number of intramolecular H-bonds to 2, 1 or 0 full bonds. The effect of partial H-bonds was tested and found to be negligible, similar to $C_5H_{12}O_6$. The results of these calculations are shown in Table 4. As expected, higher values of the c1 parameter lead to lower $p_{sat}$ values, but neither of the scaling factors obtained for $C_5H_{12}O_6$ leads to particularly good agreement with experiments for $C_5H_{10}O_6$. With c1 set to 1.61, the predicted saturation vapor pressures are significantly smaller than the experimental value. This demonstrates

the system-specific nature of the "optimal" scaling parameters, and indicates that modifying H-bonding parameters in COSMOTherm is likely not a reliable approach for estimating properties of autoxidation products. For some (stereo)isomers, restricting the number of full H-bonds to 2 or 1 had only a minor effect on $p_{sat}$, while for others a decrease of up to a factor of 5 was observed. In contrast, a dramatic decrease of $p_{sat}$ was observed for all (stereo)isomers when the number of H-bonds was restricted to zero. With this setting, the vapour pressures were all within an order of magnitude of the experimental value, and

for both diastereomers of isomer 3 even slightly below it.



### 4.Atmospheric Implications and Conclusions

Our results confirm previous suggestions that COSMOTherm tends to overestimate the saturation vapor pressures of atmospheric oxidation products capable of forming multiple intramolecular H-bonds (Kurtén et al. 2016, Krieger et al 2018).

Based on the limited number of direct comparisons between predicted and experimental data, and further assuming that the effect is roughly proportional to the maximum number of intramolecular H-bonds, the overestimation seems to be on the order of a factor of 5 per H-bond. If applied to the data in Kurtén et al. (2016) on monoterpene autoxidation products, this would imply that the originally suggested approach of taking the geometric average of SIMPOL and COSMOTherm saturation vapor pressure predictions remains a reasonably good choice in the absence of other information. While COSMOTherm saturation

vapour pressure predictions can be modified to agree better with experiments by scaling the hydrogen bond enthalpy parameter $c_1$, the optimal value for this parameter is likely to depend on the system, and this approach can thus not be generally recommended. In contrast, computing saturation vapour pressures using only conformers with a limited number of hydrogen bonds (e.g. the smallest number possible for a given system) is likely to lead to more systematic improvements, and can recommended as a temporary solution until more accurate COSMO-RS H-bonding parameterizations become available.

For users of COSMOTherm and/or COSMOconf, our results suggest that extra caution is warranted when carrying out configurational sampling of flexible molecules, especially compounds with multiple peroxy groups capable of strong intramolecular H-bonding, and possessing a large number of low-energy conformers. While the default sampling approaches may sometimes lead to better agreement with experimental results (e.g. due to missing low-energy conformers with maximal numbers of intramolecular H-bonds), they also introduce a large potential error source, as the results are heavily dependent on

the arbitrary input structure used to initiate the sampling. When computational resources permit, we therefore recommend performing systematic configurational sampling (for example using the approach described in section 2), and correcting the results where necessary, for example by selecting only conformers with a limited number of H-bonds from a sufficiently large overall set of conformers.

Finally, our results demonstrate that diastereomers of polyfunctional atmospheric oxidation products may have

different intramolecular H-bonding patterns, and thus potentially different saturation vapor pressures, solubilities and activity coefficients. While the real differences are very likely smaller than those predicted by COSMOTherm (at least in the absence of corrections for the overestimation of H-bonds), they may still be large enough to matter in the atmosphere. Stereoisomery should thus be kept in mind as a possible explanation in cases where nominally identical chemical compounds are observed to display different behaviour, for example with respect to condensation and aerosol formation.



**Supplement**

Additional technical details on sampling and selection of conformers and figures illustrating vapor pressures for isomers 2-6 of $C_5H_{12}O_6$ are available as a pdf file. COSMOTherm input files (.cosmo and .energy files) for all studied systems are provided in a zip file archived at: https://doi.org/10.5281/zenodo.1344890.

**Author contributions**

TK planned and supervised the study, with support from JT and NLP. NH, ELD and TK performed the calculations, with support from NLP. TK wrote the paper with contributions from all co-authors.

**Acknowledgments**

TK, NH and ELD thank the Academy of Finland for funding. ELD was supported by the National Science Foundation Graduate
Research Fellowship under Grant No. DGE-1256082. NLP thanks the European Research Council under the European Union's Horizon 2020 research and innovation programme (Project SURFACE, Grant Agreement No. 717022) and the Academy of Finland (Grant 308238) for funding. We thank the CSC – IT Center for Science, Finland, for computational resources and Dr. Frank Eckert from COSMOLogic GmbH for technical support and helpful discussions. We thank B.H. Lee and F.D. Lopez-Hilfiker (UW), and J. Shilling and J. Liu (PNNL) for their contributions to the observations.

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





**Table 1 – Saturation vapor pressures, in mbar, at 298.15K, of different structural and stereoisomers of the dihydroxy dihydroperoxide $C_5H_{12}O_6$, calculated using COSMOTherm version 18 and the BP_TZVPD_FINE_18 parametrization (based on BP/def2-TZVPD//BP/def-TZVP quantum chemical data). The experimental value (for an unknown mixture of isomers) is $9.87 \times 10^{-7}$ mbar.**

| Isomer | | SMILES | Best | Worst | Systematic |
|---|---|---|---|---|---|
| iso1 | R,R | $7.77 \times 10^{-5}$ | $1.38 \times 10^{-4}$ | $1.86 \times 10^{-5}$ | $1.34 \times 10^{-3}$ |
| | S,S | $7.30 \times 10^{-5}$ | $1.55 \times 10^{-4}$ | $1.76 \times 10^{-4}$ | $1.39 \times 10^{-3}$ |
| | R,S | $2.02 \times 10^{-6}$ | $4.33 \times 10^{-6}$ | $1.15 \times 10^{-4}$ | $8.82 \times 10^{-5}$ |
| | S,R | $6.68 \times 10^{-6}$ | $5.79 \times 10^{-6}$ | $2.07 \times 10^{-6}$ | $8.59 \times 10^{-5}$ |
| iso2 | R,R | $2.44 \times 10^{-4}$ | $1.77 \times 10^{-3}$ | $1.02 \times 10^{-4}$ | $9.59 \times 10^{-4}$ |
| | S,S | $3.13 \times 10^{-5}$ | $1.94 \times 10^{-3}$ | $5.42 \times 10^{-5}$ | $9.36 \times 10^{-4}$ |
| | R,S | $1.17 \times 10^{-5}$ | $2.15 \times 10^{-3}$ | $1.32 \times 10^{-5}$ | $1.46 \times 10^{-3}$ |
| | S,R | $1.34 \times 10^{-5}$ | $2.01 \times 10^{-3}$ | $1.83 \times 10^{-3}$ | $1.30 \times 10^{-3}$ |
| iso3 | R,R | $3.81 \times 10^{-5}$ | $2.18 \times 10^{-5}$ | $1.36 \times 10^{-5}$ | $1.71 \times 10^{-4}$ |
| | S,S | $4.28 \times 10^{-5}$ | $2.37 \times 10^{-5}$ | $3.94 \times 10^{-4}$ | $1.66 \times 10^{-4}$ |
| | R,S | $2.37 \times 10^{-6}$ | $1.09 \times 10^{-6}$ | $6.21 \times 10^{-5}$ | $2.59 \times 10^{-5}$ |
| | S,R | $3.62 \times 10^{-6}$ | $2.99 \times 10^{-6}$ | $8.57 \times 10^{-6}$ | $2.50 \times 10^{-5}$ |
| iso4 | S,S | $3.70 \times 10^{-5}$ | $2.04 \times 10^{-3}$ | $4.33 \times 10^{-5}$ | $1.31 \times 10^{-3}$ |
| | S,R | $2.90 \times 10^{-6}$ | $7.82 \times 10^{-4}$ | $7.60 \times 10^{-4}$ | $3.70 \times 10^{-4}$ |
| iso5 | S,S | $6.33 \times 10^{-5}$ | $1.36 \times 10^{-5}$ | $4.40 \times 10^{-5}$ | $6.64 \times 10^{-5}$ |
| | S,R | $1.64 \times 10^{-5}$ | $3.77 \times 10^{-5}$ | $1.77 \times 10^{-5}$ | $3.89 \times 10^{-5}$ |
| iso6 | R,R | $1.08 \times 10^{-3}$ | $7.24 \times 10^{-4}$ | $4.03 \times 10^{-4}$ | $4.52 \times 10^{-4}$ |
| | R,S | $1.26 \times 10^{-5}$ | $1.97 \times 10^{-5}$ | $4.95 \times 10^{-6}$ | $1.82 \times 10^{-5}$ |



**Table 2. Activities of ISOP(OOH)$_2$ species computed in water, and in a water – insoluble organic phase model (WIOM) compound "Kalberer molecule B" (Kalberer et al., 2004), at infinite dilution, and 298.15 K. Calculated using COSMOtherm version 18 and the BP_TZVPD_FINE_18 parametrization (based on BP/def2-TZVPD//BP/def-TZVP quantum chemical data).**

| Isomer | | $\gamma_{H2O}$ | $\gamma_{WIOM}$ |
|---|---|---|---|
| iso1 | R,R | 1.89 | 9.28 |
| | S,S | 1.83 | 9.82 |
| | R,S | 2.22 | 15.65 |
| | S,R | 2.27 | 16.03 |
| iso2 | R,R | 1.37 | 7.51 |
| | S,S | 1.34 | 7.74 |
| | R,S | 2.28 | 7.85 |
| | S,R | 2.33 | 8.04 |
| iso3 | R,R | 1.84 | 10.70 |
| | S,S | 1.93 | 10.71 |
| | R,S | 1.97 | 10.20 |
| | S,R | 1.94 | 10.11 |
| iso4 | S,S | 2.45 | 8.65 |
| | S,R | 2.90 | 11.49 |
| iso5 | S,S | 1.56 | 13.19 |
| | S,R | 1.34 | 12.79 |
| iso6 | R,R | 2.85 | 6.97 |
| | R,S | 1.56 | 10.51 |



**Table 3. Saturation vapor pressures and activity coefficients, at 298.15 K, of $C_5H_{10}O_6$ structural and stereoisomers, computed using COSMOTherm and the BP_TZVPD_FINE_C30_18 parametrization (based on BP/def2-TZVPD//BP/def-TZVP quantum chemical data), with the "systematic" conformer sampling scheme.**

| Isomer | | $p_{sat}$, mbar | $\gamma_{H2O}$ | $\gamma_{WIOM}$ |
|---|---|---|---|---|
| Isomer 1 | S,S | 0.00095 | 5.7 | 2.5 |
| | S,R | 0.00058 | 2.5 | 4.2 |
| Isomer 2 | R,R | 0.0010 | 4.3 | 3.2 |
| | R,S | 0.00036 | 2.8 | 2.6 |
| Isomer 3 | R,R | 0.00056 | 8.5 | 3.3 |
| | R,S | 0.00034 | 7.6 | 2.9 |





**Table 4. Saturation vapor pressures and activity coefficients, at 298.15 K, of $C_5H_{10}O_6$ structural and stereoisomers, computed using COSMOTherm and the BP_TZVPD_FINE_18 parametrization (based on BP/def2-TZVPD//BP/def-TZVP quantum chemical data), with the "systematic" conformer sampling scheme, with either modified values of the H-bonding enthalpy parameter c1, or using only conformers with a restricted number of full intramolecular H-bonds bonds.**

| Isomer | | $p_{sat}$, mbar, c1=1.36 | $p_{sat}$, mbar, c1=1.61 | $p_{sat}$, mbar, only conformers with 2 full and 0 partial H-bonds | $p_{sat}$, mbar, only conformers with 1 full and 0 partial H-bonds | $p_{sat}$, mbar, only conformers with 0 full and 0 partial H-bonds |
|---|---|---|---|---|---|---|
| Isomer 1 | S,S | $3.11\times10^{-5}$ | $1.66\times10^{-6}$ | 0.00094 | 0.00097 | $4.46\times10^{-5}$ |
| | S,R | $1.94\times10^{-5}$ | $9.83\times10^{-7}$ | 0.00057 | 0.00027 | $1.61\times10^{-5}$ |
| Isomer 2 | R,R | $2.35\times10^{-5}$ | $1.06\times10^{-6}$ | 0.00098 | 0.00027 | $5.82\times10^{-5}$ |
| | R,S | $1.00\times10^{-5}$ | $4.44\times10^{-7}$ | 0.00035 | 0.00035 | $6.05\times10^{-5}$ |
| Isomer 3 | R,R | $4.22\times10^{-5}$ | $4.60\times10^{-6}$ | 0.00055 | 0.00011 | $5.64\times10^{-6}$ |
| | R,S | $2.59\times10^{-5}$ | $2.66\times10^{-6}$ | 0.00033 | 0.00011 | $6.04\times10^{-6}$ |





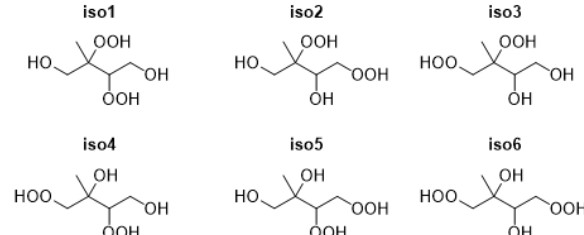

**Figure 1. Structural isomers of ISOP(OOH)$_2$.**





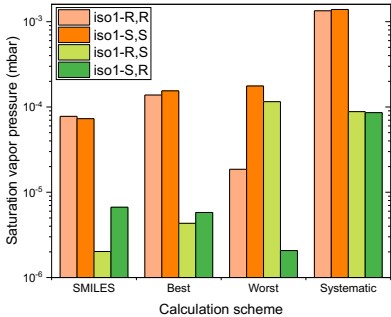

**Figure 2. Saturation vapor pressures, at 298.15 K, of different stereoisomers of structural isomer 1 of the dihydroxy dihydroperoxide C₅H₁₂O₆, at 298.15K, calculated using COSMOtherm version 18 and the BP_TZVPD_FINE_18 parametrization (based on BP/def2-**
5   **TZVPD//BP/def-TZVP quantum chemical data), using different conformational sampling schemes.**





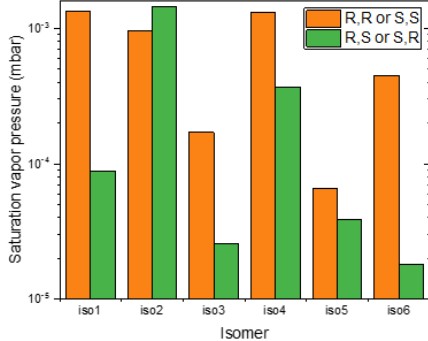

**Figure 3. Saturation vapor pressures, at 298.15 K, of different stereoisomers of structural isomer 1 of the dihydroxy dihydroperoxide**
5 **$C_5H_{12}O_6$, at 298.15K, calculated using COSMOtherm version 18 and the BP_TZVPD_FINE_18 parametrization (based on BP/def2-TZVPD//BP/def-TZVP quantum chemical data), with the systematic conformational sampling scheme.**





iso1-S,S      iso1-S,R

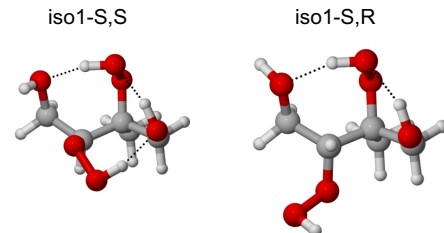

**Figure 4. Lowest-energy conformers (at the BP/def2-TZVPD//BP/def-TZVP level, with COSMO solvation, using the systematic configurational sampling scheme) of the S,S (left) and S,R (right) stereoisomers of structural isomer 1 of ISO(POOH)$_2$. The S,R stereoisomer only has two intramolecular H-bonds in the lowest-energy conformer while the S,S isomer has three. Correspondingly, the saturation vapor pressure of the S,R stereoisomer is predicted to be a factor of 15 lower. Color coding: gray=C, red=O, white=H.**




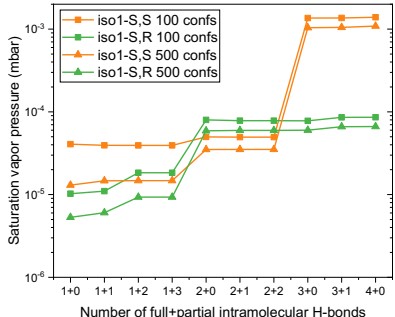

**Figure 5. Saturation vapor pressures, at 298.15 K, of the S,S and S,R stereoisomers of structural isomer 1 of the dihydroxy dihydroperoxide $C_5H_{12}O_6$, calculated using COSMOtherm version 18 and the BP_TZVPD_FINE_18 parametrization (based on BP/def2-TZVPD//BP/def-TZVP quantum chemical data), including only conformers with less than *n* + *m* full and partial H-bonds (where e.g. "1+3" indicates one full and three partial H-bonds), and using either 100 or 500 conformers in the original conformer pool.**




**Figure 6.** $C_5H_{10}O_6$ isomers and stereoisomers used in the calculations. **Top row: isomer 1. Middle row: isomer 2. Bottom row: isomer 3. The chirality of stereocenters are indicated with "S" and "R" letters.**

