# Peer review of "Estimating the saturation vapor pressures of isoprene oxidation products $C_5H_{12}O_6$ and $C_5H_{10}O_6$ using COSMO-RS"

_Atmospheric Chemistry and Physics, 2018_

## Referee Comment (RC1) · Anonymous Referee #1 · 28 Sep 2018

This is highly technical paper that provides some important insights into the capability of a method based on quantum chemistry and statistical thermodynamics to accurately predict the saturation vapor pressure of small organic substances with a very large number of hydrogen-bond forming functional groups (hydroxy, peroxy). The relevance of the work derives from the importance of such compounds as SOA forming oxidation products of isoprene, and the fact that such substances clearly fall far outside of the applicability domain of group contribution-based prediction methods commonly used in the SOA community. The paper convincingly demonstrates that the prediction method greatly overestimates the impact of intramolecular hydrogen bonding on saturation vapor pressure, which leads to overprediction by as much as three orders of

magnitude. It also explores several approaches to addressing this issue and makes recommendation on how more realistic estimates could be obtained. Considering the highly technical nature of the material, the paper is well written and also accessible to those not intimately familiar with theoretical chemistry.

I have little to criticize but feel strongly about the fact that the paper uses non-SI units for pressure and energy. The guidelines for authors of manuscript submitted to ACP explicitly states: "The metric system is mandatory and, wherever possible, SI units should be used." Please, no lame excuses, such as "it is customary in the theoretical chemistry community to use calories" or "the measured vapor pressures we use for comparison were reported in mbar". There is absolutely no reason why it is not possible to use Pascal for pressure and Joule for energy. The unit of Pascal for pressure was introduced in 1971, i.e., well before most, if not all, of the authors of this manuscript were born!

Page 3, line 26 and 27: use Pascal

Page 8, line 25: use kJoules

Page 18: All of the data in Table 1 should be reported in units of Pascal

Page 20: The vapor pressures in Table 3 should be reported in units of Pascal

Page 21: All of the data in Table 4 should be reported in units of Pascal (The table caption suggests that Table 4 also report activity coefficients, which does not appear to be the case)

Page 23, 24 and 26: Figures 2, 3 and 5: the y-axis should have values reported in units of Pascal

The name of the software is COSMOtherm and not COSMOTherm. This should be changed throughout the manuscript.

Page 2, line 28: "catechol" instead of "cathecol"

Page 3, line 15: "nor" instead of "or"

Page 3, line 30: "up to four" instead of "up four"

Page 11, line 14: "can be recommended" instead of "can recommended"

Page 13, Line 15: "EVAPORATION" instead of "EVAPO -RATION"

---

## Referee Comment (RC2) · Anonymous Referee #2 · 16 Oct 2018

Atmospheric oxidation products are usually multifunctional. Prediction of their vapor pressures has large uncertainties due to the complicated intramolecular interactions of these molecules. In this paper, the authors compared predicted vapor pressures with experimentally measured values. Thorough comparison using different methods to derive conformers for the calculation of vapor pressure based on COSMO-RS was conducted. These comparisons suggest the possibility of overestimation of saturation vapor pressures due to the overestimation of intramolecular interactions, specifically H-bond, by COSMO-RS. Although in this study, the authors have only selected two isoprene oxidation products which might be due to high computation cost, this method may be applicable to other atmospherically relevant molecules. The results of this study

suggest the importance of proper treatment of intramolecular interaction and selection of conformers in predicting vapor pressures for multifunctional atmospheric oxidation products. Overall, the description of the method used for the calculation in the paper is clear and well written. The results and discussion are a useful contribution to the literature on the estimation of vapor pressures for multifunctional organic compounds. I have a few minor comments.

Page 7 line 29: The high activity coefficients in WIOM phase (and higher than in pure water) suggests that WIOM phase used in this study may be not a good representative organic phase for isoprene SOA as measured by D'Ambro et al. (2007). Isoprene SOA is much more polar than WIOM phase. This probably won't change the conclusion that "Differences between the measured values and true pure-compound saturation vapor pressures can thus not explain the discrepancy between COSMOTherm results and measurements."

Page 9 line 2: It is not clear why the authors varied $c_1$ between 0.6 and 1.8. Are 0.6 and 1.8 threshold values for $c_1$?

---

## Author Comment (AC1) · 15 Nov 2018

We thank both reviewers for their constructive comments to our manuscript "Estimating the saturation vapor pressures of isoprene oxidation products $C_5H_{12}O_6$ and $C_5H_{10}O_6$ using COSMO-RS". We have revised the manuscript accordingly.

We have reproduced below the reviewer comments (*in italics*), our response (**in bold**) and the changes we have made to the manuscript (underlined).

**Reviewer 1:**

*This is highly technical paper that provides some important insights into the capability of a method based on quantum chemistry and statistical thermodynamics to accurately predict the saturation vapor pressure of small organic substances with a very large number of hydrogen-bond forming functional groups (hydroxy, peroxy). The relevance of the work derives from the importance of such compounds as SOA forming oxidation products of isoprene, and the fact that such substances clearly fall far outside of the applicability domain of group contribution-based prediction methods commonly used in the SOA community. The paper convincingly demonstrates that the prediction method greatly overestimates the impact of intramolecular hydrogen bonding on saturation vapor pressure, which leads to overprediction by as much as three orders of magnitude. It also explores several approaches to addressing this issue and makes recommendation on how more realistic estimates could be obtained. Considering the highly technical nature of the material, the paper is well written and also accessible to those not intimately familiar with theoretical chemistry.*

*I have little to criticize but feel strongly about the fact that the paper uses non-SI units for pressure and energy. The guidelines for authors of manuscript submitted to ACP explicitly states: "The metric system is mandatory and, wherever possible, SI units should be used." Please, no lame excuses, such as "it is customary in the theoretical chemistry community to use calories" or "the measured vapor pressures we use for comparison were reported in mbar". There is absolutely no reason why it is not possible to use Pascal for pressure and Joule for energy. The unit of Pascal for pressure was introduced in 1971, i.e., well before most, if not all, of the authors of this manuscript were born!*
*Page 3, line 26 and 27: use Pascal*
*Page 8, line 25: use kJoules*
*Page 18: All of the data in Table 1 should be reported in units of Pascal*
*Page 20: The vapor pressures in Table 3 should be reported in units of Pascal*
*Page 21: All of the data in Table 4 should be reported in units of Pascal (The table caption suggests that Table 4 also report activity coefficients, which does not appear to be the case)*
*Page 23, 24 and 26: Figures 2, 3 and 5: the y-axis should have values reported in units of Pascal*
*The name of the software is COSMOtherm and not COSMOTherm. This should be changed throughout the manuscript.*
*Page 2, line 28: "catechol" instead of "cathecol" C2*
*Page 3, line 15: "nor" instead of "or"*
*Page 3, line 30: "up to four" instead of "up four"*
*Page 11, line 14: "can be recommended" instead of "can recommended" Page 13, Line 15: "EVAPORATION" instead of "EVAPO -RATION"*

**Response**: We thank the reviewer for his or her constructive comments. As our lame excuses concerning traditional units have been pre-empted, we have followed the reviewer's recommendations and replaced all non-SI units by SI units (both in the text, in Tables 1, 3 and 4, in the Figures, and also in the tables and figures of the supplemental information file). We have also removed mention of activity coefficients from the caption of Table 4, and used the correct capitalisation for COSMOtherm. The other spelling mistakes noted have also been corrected.

**Changes made to the manuscript (page and line numbers correspond to revised manuscript):**

-COSMOTherm replaced by COSMOtherm throughout the manuscript.

-Mbar changed to Pa and kcal/mol to kJ/mol throughout the manuscript (and the SI file).

Spelling errors corrected as suggested:

-Page 2, line 27: "cathecol" replaced by "catechol"
-Page 3, line 14: "or" replaced by "nor"
-Page 3, line 30: "up four" replaced by "up to four"
-Page 11, line 14:  "can recommended" replaced by "can be recommended"
-Page 13, Line 15: "EVAPO -RATION" replaced by "EVAPORATION"

**Reviewer 2:**

*Atmospheric oxidation products are usually multifunctional. Prediction of their vapor pressures has large uncertainties due to the complicated intramolecular interactions of these molecules. In this paper, the authors compared predicted vapor pressures with experimentally measured values. Thorough comparison using different methods to derive conformers for the calculation of vapor pressure based on COSMO-RS was conducted. These comparisons suggest the possibility of overestimation of saturation vapor pressures due to the overestimation of intramolecular interactions, specifically H-bond, by COSMO-RS. Although in this study, the authors have only selected two isoprene oxidation products which might be due to high computation cost, this method may be applicable to other atmospherically relevant molecules. The results of this study suggest the importance of proper treatment of intramolecular interaction and selection of conformers in predicting vapor pressures for multifunctional atmospheric oxidation products. Overall, the description of the method used for the calculation in the paper is clear and well written. The results and discussion are a useful contribution to the literature on the estimation of vapor pressures for multifunctional organic compounds. I have a few minor comments.*
*Page 7 line 29: The high activity coefficients in WIOM phase (and higher than in pure water) suggests that WIOM phase used in this study may be not a good representative organic phase for isoprene SOA as measured by D'Ambro et al. (2007). Isoprene SOA is much more polar than WIOM phase. This probably won't change the conclusion that "Differences between the measured values and true pure-compound saturation vapor pressures can thus not explain the discrepancy between COSMOTherm results and measurements."*
*Page 9 line 2: It is not clear why the authors varied $c_1$ between 0.6 and 1.8. Are 0.6 and 1.8 threshold values for $c_1$?*

**Response: We thank the reviewer for his or her constructive comments. The choice of two isoprene oxidation products was governed partially by computational cost (while calculations using the default COSMOtherm/COSMOconf approach are cheap, the full conformational sampling carried out here can become expensive), but the main reason was that there are very few published experimental saturation vapor pressure values for compounds containing more than one OOH group: the two investigated here were the only ones available at the time of study. We agree with the reviewer that the isoprene SOA is likely to be more polar than the model WIOM, and have added mention of this to the manuscript. The limits of 0.6 and 1.8 chosen for our initial testing of the effect of varying the $c_1$ parameter were arbitrary, our intent was to investigate what $c_1$ values are needed to match ("fit") the experimental saturation vapor pressures, and this is simply the range we started out from. (As it turned out, the "fitted" values of 1.61 and 1.36, depending on the enantiomer pair, happened to fall within this range.) This has now been clarified in the manuscript.**

**Changes made to the manuscript (page and line numbers correspond to revised manuscript):**

-Page 7, line 25, added sentence: "We note that the actual organic phase used in the experiments is likely to be more polar than this model WIOM phase."

-Page 9, line 2, "between 0.6 and 1.8" replaced by "between arbitrarily chosen limits of 0.6 and 1.8"